# The Volume of Non-Restricted Boltzmann Machines and Their Double Descent Model Complexity

## Abstract

The double descent risk phenomenon has received much interest in the machine learning and statistics community. Motivated through Rissanen's minimum description length (MDL) principle, and Amari's information geometry, we investigate how a double descent-like behavior may manifest by considering the $\log V$ modeling term - which is the logarithm of the model volume. In particular, the $\log V$ term will be studied for the general class of fully-observed statistical lattice models, of which Boltzmann machines form a subset. Ultimately, it is found that for such models the $\log V$ term can decrease with increasing model dimensionality, at a rate which appears to overwhelm the classically understood $\mathcal{O}(D)$ complexity terms of AIC and BIC. Our analysis aims to deepen the understanding of how the double descent behavior may arise in deep lattice structures, and by extension, why generalization error may not necessarily continue to grow with increasing model dimensionality.

## 1 Introduction

Model selection is a problem which underpins the field of machine learning. Key to its formulation is the notion of learning an appropriate predictor, $h^\star : \mathbb{R}^D \to \mathbb{R}$ from an underlying model class, $\mathcal{H}$, based on $N$ input training examples $\{(\boldsymbol{x}_i, y_i)\}_{i=1}^N$, with each $(\boldsymbol{x}_i, y_i) \in \mathbb{R}^D \times \mathbb{R}$. Typically, the predictor, $h^\star$, is chosen so as to minimize some risk functional; that is, $h^\star = \arg\min_{h \in \mathcal{H}} R(h)$ with $R(h) = \mathbb{E}_{p(x,y)}[L(h(x), y)]$, where $L : \mathbb{R} \times \mathbb{R} \to \mathbb{R}$ is the risk functional, and $p(x, y)$ denotes the probability density function (pdf) over the data. Fundamentally, the aim of such an approach is to ensure that $h^\star$ provides good generalization capability, so that after training it minimizes the *out-of-sample* test error [12]. This is historically estimated via the Akaike information criterion (AIC) [3], the Bayesian information criterion (BIC) [29], or through cross validation [12]. AIC and BIC are derived based on asymptotic assumptions in the sample size $N$, and work similarly. Moreover, both criteria suggest that out-of-sample error increases as $\mathcal{O}(D)$[1], suggesting that an over-parameterized model should generalize poorly, which is an idea consistent with traditional empirical evidence, via the U-shaped *train-test* curves [12].

Recently, however, particular classes of highly parameterized models such as deep neural networks, and random forests have been shown to generalize extremely well, working in contrary to the implied $\mathcal{O}(D)$ model complexity effects. In fact, strong empirical evidence has been presented by Belkin et al. [10], where it was shown that a *double descent risk* phenomenon may be observed for a variety of models which transition into highly parameterized regimes. This phenomenon is shown in Figure A2a in Appendix, with many additional experiments made clear in [25]. In an effort to explain such trends Belkin et al. [11] have tried to infer some similarities between ReLU networks and traditional

---

[1]Technically AIC has a $2D$ model complexity term, and BIC has a $D \log N$ model complexity term. We will refer to the implied effects of both model complexities simply as the "$\mathcal{O}(D)$ terms".

kernel models, and Geiger et al. [15] have tried to connect the double descent cusp-like behaviour with diverging norms, through a neural tangent kernel framework. In addition, double descent risk has been explored in a variety of simpler (and shallow) model classes [17, 6, 13], with various risk asymptotics established [9, 20]. Lastly and rather interestingly, it has been found that there exist certain parallels between double descent risk, and the notion of the *jamming transition* which occurs in physical materials which undergo a phase transition [31, 16]. In this paper we look into the problem of how increasing the underlying dimensionality of statistical lattice models (of which Boltzmann machines form a subset) could imply increasing generalizability. Such ideas are motivated from the recent empirical findings of double descent risk. This will be achieved via the notion of a *model volume*, which carries interpretations from information theory, and Occam's razor. In particular it will be shown the it is possible to have a decreasing model volume in such models as $D$ increases, which tends to offset the $\mathcal{O}(D)$ complexity terms of AIC and BIC.

## 1.1 Model Selection and Occam's Razor

In the late 90s and early 2000s, extensions to the base AIC and BIC formulations were developed by Rissanen [28] and Balasubramanian [8], which include additional model-specific terms. From the perspective of coding theory, Rissanen developed a notion of *stochastic model complexity*, which builds upon Shannon's information criteria used for lossless encoding [30]. Upon this notion, Rissanen formalized an extension of binary Shannon entropy to continuous function classes, via the discretization of the model manifold over approximately equivalent model classes. This approach establishes an intuition behind "model distinguishability", which is also echoed by Balasubramanian. In particular, under Risannen's construction information is encoded in *nats* (as opposed to bits) and it is formally recognized as the *Minimum Description Length* (MDL). This is shown in Equation (1),

$$-\log(p(\boldsymbol{x})) = \overbrace{-\log(\hat{\mathcal{L}}) + \frac{D}{2}\log\left(\frac{N}{2\pi e}\right)}^{\text{AIC / BIC - like term}} + \overbrace{\log\int_{\Theta}\sqrt{\det\left(\mathcal{I}(\boldsymbol{\theta})\right)}d\boldsymbol{\theta}}^{\text{Log - Model Volume}} + o(1), \quad (1)$$

where $\boldsymbol{x} = \{\boldsymbol{x}_i\}_{i=1}^{N}$ denotes a random vector of $N$ data samples, $\hat{\mathcal{L}} \in \mathbb{R}$, is the likelihood function evaluated at its optimal parameter setting, with $\Theta$ being the space of possible parameter settings, and $\mathcal{I}(\boldsymbol{\theta}) \in \mathbb{R}^{D \times D}$ denotes the Fisher information matrix (FIM), which is traditionally used as a lower bound on the variance of unbiased estimators, and in Jeffrey's prior [12]. In a parallel fashion, Balasubramanian approached the problem of model selection, albeit from a more Bayesian perspective, which yielded a very similar formulation to Rissanen's MDL [7]. To achieve this he takes an alternate route based on specifying a Jeffrey's prior over the underlying parameter space, which is noted to act as an appropriate measure for the density of distinguishable distributions [8]. Ultimately, in both Rissanen's and Balasubramanian's model selection criteria, there is a term which acts like the $\mathcal{O}(D)$ model complexity used in AIC and BIC, and an additional term: $\log\int_{\Theta}\sqrt{\det\left(\mathcal{I}(\boldsymbol{\theta})\right)}d\boldsymbol{\theta}$, which they collectively referred to as the model distinguishability-like term [19]. Moreover, since these methods are built upon the log-marginal: $-\log(p(\boldsymbol{x}))$, they also appeal to a Bayesian Occam's razor-like principle. Ultimately, the ensuing study will aim to explore the term: $\log\int_{\Theta}\sqrt{\det\left(\mathcal{I}(\boldsymbol{\theta})\right)}d\boldsymbol{\theta}$, for statistical lattice models. In particular, it will be made clear that this term describes the underlying log-model volume, which we denote by $\log V$, and that this term can in fact *decrease* with increasing $D$ in statistical lattice models. This behaviour is significant as it suggests that certain model classes can have the power to generalize well when transitioning into the over-parameterized regimes.

Before proceeding to analyze this volume term, it is important to clarify certain points of Equation (1). In particular, it should be noted that the $o(1)$ term is defined with respect to $N$, and thus it is *not necessarily $o(1)$* with respect to $D$. In fact, it is possible for there to exist additional terms which behave as a function of $D$. Indeed if one approaches the derivation of an MDL-like criteria through Balasubramanian's razor, the additional modeling term: $\log\sqrt{\det(\mathcal{I}(\hat{\boldsymbol{\theta}}))/\det(\tilde{\mathcal{I}}(\hat{\boldsymbol{\theta}}))}$ appears, where $\tilde{\mathcal{I}}$ denotes the empirical FIM, and $\hat{\boldsymbol{\theta}}$ denotes the value of $\boldsymbol{\theta}$ at the maximum likelihood location. This is often interpreted as an indicator of model *robustness* with respect to the location $\hat{\boldsymbol{\theta}}$ on the manifold [8]. It is also possible to derive explicit notions of geometric curvature (in particular, the Ricci-curvature tensor) in the MDL expression if one so wishes [23] - but such expressions are not central to the present study. This is because the authors are more concerned with the holistic (i.e. geometrically intrinsic) viewpoint of the underlying model space, but these model curvature terms typically require a specific $\hat{\boldsymbol{\theta}}$ value. Moreover, the geometric model volume provides an intuitive means in which to

establish notions of *model distinguishability* [7, 28, 24]. In other words, we are concerned with terms which encode intrinsic *geometric complexity*, whereas $\log \sqrt{\det(\mathcal{I}(\hat{\boldsymbol{\theta}}))/\det(\tilde{\mathcal{I}}(\hat{\boldsymbol{\theta}}))}$ is often said to model *relative complexity* [24].

## 1.2 Information Geometry

Information geometry concerns the application of differential geometry to statistical models. In particular, it considers a statistical manifold, $\mathcal{M} = \{p(x; \theta)\}$, over a $\theta$ co-ordinate system. A Riemannian metric, $\mathcal{G}$, can be placed on $\mathcal{M}$, where $\mathcal{G} : \mathcal{T}_p(\mathcal{M}) \times \mathcal{T}_p(\mathcal{M}) \to \mathbb{R}_{\geq 0}$ for each point $p \in \mathcal{M}$, with $\mathcal{T}_p(\mathcal{M})$ defined as the local tangent space at point $p$ on the manifold. Principally, $\mathcal{G}$ is a generalization of the inner product on Euclidean spaces to Riemannian manifolds. In addition, Amari defines a dually coupled affine co-ordinate system on statistical manifolds. Dually coupled co-ordinates arise naturally from the *dually flat* property which is intrinsic to many information manifolds. These co-ordinates are known as the $\theta$ (e-flat) and $\eta$ (m-flat) co-ordinates for the exponential family in particular, and are related through the Legendre transformation $\eta = \nabla \psi(\theta)$ and $\theta = \nabla \varphi(\eta)$ via two convex functions $\psi, \varphi : \mathbb{R}^D \to \mathbb{R}$ [5]. The $\theta$ and $\eta$ co-ordinates for exponential models correspond to the natural and expectation parameters, respectively. Furthermore, the FIM defines a natural Riemannian metric tensor: $\mathcal{G}_{ij} = \mathbb{E}[\partial_i \log(p(x; \theta)) \partial_j \log(p(x; \theta))] = \mathcal{I}_{ij}$ [26, 5]. Thus the motivation for using information geometry is clear as Rissanen's MDL, and Balasubramanian's Occam Razor, depend on the FIM, which is, geometrically speaking, the metric tensor. Consequently, the term $\log \int_\Theta \sqrt{\det(\mathcal{I}(\boldsymbol{\theta}))} d\boldsymbol{\theta}$ has a clear definition in differential geometry as being the log-*volume* of the underlying information manifold; that is, the square root of the determinant of the Fisher information matrix is the manifold volume [5, 21].

# 2 Statistical Lattice Models and Model Volumes

Statistical lattice models are popular, traditional machine learning models which include *Boltzmann machines* (or Ising models) [1], log-linear models [4], and the matrix balancing problem [33]. In this section, we will work the hierarchical encoding of a probability distribution via a *lattice* structure [22, 32], which will be shown to naturally lead into the $\eta$ co-ordinates ($m$-flat) from information geometry (see §1.2), and analyze the learning of distributions over a lattice structured domain.

Formally, a *partially ordered set* (*poset*) is a tuple, $(\mathcal{P}, \leq_\mathcal{P})$, where $\mathcal{P}$ is a set of elements, and $\leq_\mathcal{P}$ denotes an ordering structure, such that (1) $\forall p^2 \in \mathcal{P}$, $p \leq_\mathcal{P} p$ (reflexivity), (2) If $p \leq_\mathcal{P} q$, and $q \leq_\mathcal{P} p$, then $p = q$ (antisymmetry), and (3) If $p \leq_\mathcal{P} q$, and $q \leq_\mathcal{P} r$, then $p \leq_\mathcal{P} r$. Note that not every element may be directly comparable to every other element in the set (which would be known as a *total* ordering). In addition, a poset $(\mathcal{P}, \leq_\mathcal{P})$ is called a *lattice* if every pair of elements $p, q \in \mathcal{P}$ has the least upper bound $p \vee q$ and the greatest lower bound $p \wedge q$ [14]. We assume that $\mathcal{P}$ is finite. In working with posets it is common to consider the zeta function, $\zeta : \mathcal{P} \times \mathcal{P} \to \{0, 1\}$ such that $\zeta(p, q) = \mathbf{1}_{p \leq q}$ [18]. The lattice structure always gives us the $\theta$ and $\eta$ co-ordinates of a statistical manifold: $\log \mathbb{P}(p) = \sum_{q \in \mathcal{P}} \zeta(q, p) \theta_q = \sum_{q \leq p} \theta_q$ and $\eta_p = \sum_{q \in \mathcal{P}} = \zeta(p, q) \mathbb{P}(q) = \sum_{q \geq p} \mathbb{P}(q)$ [33]. For example, for Boltzmann machines with $d$ binary variables, the lattice space $\mathcal{P} = \{0, 1\}^n$, where $p = (p_1, \ldots, p_n) \leq_\mathcal{P} q = (q_1, \ldots, q_n)$ if $p_i \leq q_i$ for all $i \in \{1, \ldots, n\}$. The size $D = |\mathcal{P}| = 2^n$ in this case. The metric tensor for the information manifold of the proposed lattice structure is shown in Theorem 1, which was previously derived by Sugiyama et al. [33]. We assume that $\mathcal{P} = \{1, \ldots, |\mathcal{P}|\}$ such that 1 corresponds to the least element without loss of generality.

**Theorem 1 (Lattice Metric Tensor [33]).** $\mathcal{G}_{ij} = \sum_{p \in \mathcal{P}} \zeta(i, p) \zeta(j, p) \mathbb{P}(p) - \eta_i \eta_j = \eta_{i \vee j} - \eta_i \eta_j$.

In Theorem 1, we can replace $\sum_{p \in \mathcal{P}} \zeta(i, p) \zeta(j, p) \mathbb{P}(p)$ with $\eta_{i \vee j}$ as we assume that $\mathcal{P}$ is a lattice and $\eta_{i \vee j}$ always exists, which says that we only consider those poset structures in which the $\eta$ co-ordinate values are shared (nested) between $\eta_i$ and $\eta_j$ for the off-digagonal terms in the metric tensor. In Theorem 1 it is clear that this metric tensor is expressed in terms of the $\eta$ co-ordinates. The equivalent metric tensor in terms of the $\theta$ co-ordinates is available, but it is much more difficult to work with (requires the Möbius function instead of the zeta function). Moreover, in this definition of the metric tensor, the first row and column are always zeros, resulting in again, a singular geometry. Luckily this

---

[2]Since elements from a poset are denoted by $p$, we denote a probability measure as $\mathbb{P}$ when referencing poset systems.

time however, since $-\theta_1$ corresponds to the partition function, it is generally removed in practice, [34] so that we effectively work with co-ordinates $\boldsymbol{\eta}' = (\eta_2, \ldots, \eta_D)$, resulting in $\mathcal{G}' \succ 0$.

Based on this set-up, it is possible to derive the upper and lower bounds for $\log V$. For lattice models, the $\eta$ co-ordinates lie compactly on a simplex within the unit $D$-hypercube, that is $\Omega = [0, 1]^D$, which makes evaluations much simpler [32]. In fact, it is possible to perform the re-parameterization: $\boldsymbol{\delta} = f(\boldsymbol{\eta})$, which allows $\boldsymbol{\delta}$ to be sampled from a Dirichlet distribution. This makes the $\eta$ co-ordinate more intuitive to work with, and provides us with a tractable way to evaluate the volume integral via sampling. We provide details of this re-parameterization in Appendix A.1. The $\log V$ bounds which result are shown in Theorem 2, with the proof clarified in Appendix A.2.

**Theorem 2 (Lattice Log Volume Bounds).** $\log V$ *is bound as in Equation* (2)*, where* $\mathcal{G} = \mathcal{M}^{\mathsf{T}}\mathcal{M}$, $\boldsymbol{\delta} = f(\boldsymbol{\eta})$ *is a re-paramterization, and* $\Gamma(D) = (D - 1)!$ *is the standard Gamma function.*

$$
\overbrace{\mathbb{E}\left[\sum_{i=1}^{D} \log\left(\mathcal{M}_{ii}(\boldsymbol{\delta})\right)\right]}^{\text{``Richness''}} + \overbrace{\log\left(\frac{1}{\Gamma(D)}\right)}^{\text{``Distinguishability''}} \leq \log V \leq \log\left(\overbrace{\mathbb{E}\left[\sqrt{\prod_{i=1}^{D} \mathcal{G}_{ii}(\boldsymbol{\delta})}\right]}^{\text{``Richness''}}\right) + \overbrace{\log\left(\frac{1}{\Gamma(D)}\right)}^{\text{``Distinguishability''}} \quad (2)
$$

As Theorem 2 makes clear, the $\log V$ term can be decomposed into two components which we define as: (i) *Model richness*: which is driven by the elements found in the metric tensor, and (ii) *Model distinguishability*: which refers to the volume of a *probability simplex*. Intuitively, (i) For (higher-order) Boltzmann machines, multi-way interactions can be encoded in a desired lattice structure, and this in turn will drive the construction of the metric tensor via the $\eta$ co-ordinate system (Theorem 1). Thus, since the metric tensor depends strongly upon the chosen lattice, and since it is used in defining angles and geodesics over a manifold, it would appear that this expression encodes a notion of *model richness* and or expressibility over the manifold. As for point (ii), since the $\eta$ co-ordinate system is constrained to lie on a simplex geometry (as made explicit by the poset structuring) the volume is: $1/\Gamma(D) = 1/(D - 1)!$. Evidently, as dimensionality increases the simplex volume decreases. A nice combinatorial intuition of this is that for $D$ randomly sampled numbers $\{n_i\}_i^D$, the probability of obtaining a permutation which is precisely the total ordering of these $D$ points (i.e. $n_1 < n_2 < ... < n_D$) is $1/D!$. Thus, even if the AIC $\mathcal{O}(D)$ model complexity term grows without bound, the simplex constraint over the $D$ parameters serves to act as a strong counter-balance. In fact, this counter-balancing imbues the MDL expression with a *double descent*-like behaviour. As $D$ increases, the value of the MDL expression first increases, and then decreases. Since MDL relates strongly to the notion of model generalizbility the double descent phenomenon which arises here paints an intriguing picture for the double descent risk phenomenon that has empirically arisen in the deep learning field. A visual example of this on toy values is made clear in Figure A2b. It is important to note that our analysis assumes access to a *fully observable* lattice model (and thus is not immediately applicable to restricted Boltzmann machines without careful consideration). This is because in latent hierarchical settings the underlying geometry can become geometrically singular, which complicates the global model volume calculation with respect to the FIM [36**?** , 35].

In regards to (ii) model distinguishability, it should be noted that the term *distinguishability* is somewhat overloaded here, since in classic MDL literature it has been traditionally reserved for the entire expression: $\int_{\boldsymbol{\Theta}} \sqrt{\det(\mathcal{I}(\boldsymbol{\theta}))} d\boldsymbol{\theta}$ [8, 28]. Our proposal is to elaborate it slightly by splitting this term into two terms, where one term works to clarify the importance of the model architecture, and the other term clarifies the constraints that exist in the underlying parameter space.

Inspecting these terms in the limit of the over-parameterized regime results in the following (see Appendix A.3 for its proof).

**Remark 1 (Limiting Lattice Volume).** $\lim_{D \to \infty} V = 0$.

Thus, the volume of the proposed statistical lattice models tends towards zero for large $D$, and this limit can be said to converge factorially due to the simplex volume. Finally, it is interesting (and tempting) to relate the volume calculations established here to the model volumes of tree structures, particularly since these models have similar graphical topologies to statistical lattices. Doing so, one is able to establish Remark 2, in which it can be seen that two topologically similar models do indeed consider similar volume expressions, and that in both expressions $\lim_{D \to \infty} V = 0$ holds.

**Remark 2 (Rissanen's Tree Volume [28]).** *Binary decision trees for encoding $D$-classes have log-volume:* $\log_2 V = D/2 \log_2 \pi - \log_2 \Gamma(D/2)$. *Ignoring higher order terms, statistical lattice structures have their log-volume upper-bounded as:* $\log_e V \leq D/2 \log_e \pi - \log_e \Gamma(D/2)$.

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

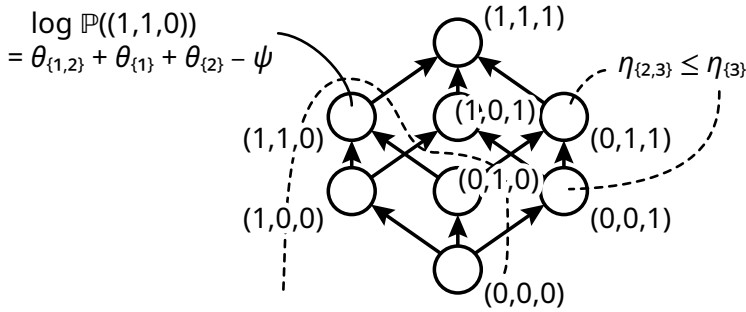

Figure A1: An example of a lattice structure for the binary domain $\{0,1\}^3$. Each arrow denotes the partial order between elements in the lattice.

## Appendix

## A  Statistical Lattice Models

### A.1  Reparameterzing the Dual Geometry of Lattices

Lattices are useful structures, as they allow one to efficiently encode information hierarchically. A geometry over lattice-type structures based on modelling higher order feature interactions via log probabilities has been derived in the work of Sugiyama et al. [32]. The well-known log-linear model for binary variables in question is formulated as,

$$\log \mathbb{P}(\boldsymbol{x}) = \sum_i \theta_{\{i\}} x_i + \sum_{i<j} \theta_{\{i,j\}} x_i x_j + \sum_{i<j<k} \theta_{\{i,j,k\}} x_i x_j x_k + \cdots + \theta_{\{1,\ldots,n\}} x_1 \ldots x_n - \psi,$$

where $\boldsymbol{x} \in \{0,1\}^n$, each $\theta \in \mathbb{R}$ denotes the connection strength of a particular higher order interaction, each $x_i \in \{0,1\}$ denotes a binary valued variable which activates a particular connection strength, $\psi \in \mathbb{R}$ denotes the normalization constant for the probability model [4, 2]. Under this structure, $\mathbb{P}(\boldsymbol{x})$ is a member of the exponential family of distributions. If we define a particular instance of the partial ordering as $\boldsymbol{x} = (x_1, \ldots, x_n) \leq \boldsymbol{y} = (y_1, \ldots, y_n)$, where $x_i \leq y_i$ for all $i \in \{1, \ldots, n\}$, and denote by $\Sigma(\boldsymbol{x})$ as the set of indices of "1" in $\boldsymbol{x}$, then when can instantiate a lattice, and can condense the representation of the above log-linear model as:

$$\log \mathbb{P}(\boldsymbol{x}) = \sum_{\boldsymbol{s}} \delta(\boldsymbol{s}, \boldsymbol{x}) \theta_{\Sigma(\boldsymbol{s})} = \sum_{\boldsymbol{s} \leq \boldsymbol{x}} \theta_{\Sigma(\boldsymbol{s})}, \quad \text{where } \psi = -\theta_\emptyset.$$

Hence the lattice is a natural representation of this hierarchical structure over the sample space of $\{0,1\}^n$. Sugiyama et al. [32] studied geometric structure of statistical lattice models and showed that distributions over not only $\{0,1\}^n$ but any lattices belong to the exponential family. Note that posets are originally used in [32], which is a more general structure than lattices. Although we treat only lattices in this paper, most interesting statistical models (such as Boltzmann machines) are lattices. We thus proceed in this direction as lattice structures entail a simple co-ordinate representation of the metric tensor as we have described in Theorem 1. An example of a lattice structure for $\{0,1\}^3$ is illustrated in Figure A1.

As Amari notes [5], the exponential family of distributions induces a statistical manifold which possesses an interesting dualisitc structure. That is, two co-ordinate systems can be dually connected and allow one to generalize notions such as the Pythagoras theorem in Euclidean manifolds, to more general statistical manifolds. In particular, for the exponential family the first of these co-ordinate systems is given by $\boldsymbol{\theta} = (\theta_1, ..., \theta_D)$ (as defined in the specified log-linear model of this subsection), and the second is given by $\boldsymbol{\eta} = (\eta_1, ..., \eta_D)$. Note that in the case of a binary log-linear model, we have $D = 2^n$ and

$$\eta_{\{i\}} = \mathbb{E}[x_i] = \Pr(x_i = 1)$$
$$\eta_{\{i,j\}} = \mathbb{E}[x_i x_j] = \Pr(x_i = 1, x_j = 1)$$
$$\eta_{\{1,\ldots,n\}} = \mathbb{E}[x_1 ... x_n] = \Pr(x_1 = 1, ..., x_n = 1),$$

and $(\boldsymbol{\theta}, \boldsymbol{\eta})$ are explicitly dually connected via the Legendre transformation [5, 32]. Note that since the $\eta$ co-ordinate system is defined probabilistically, and is thus constrained to be in $[0, 1]^D$, which is convenient, and simplifies many calculations. Moreover, note that the $\eta$ co-ordinate system is built hierarchically, in the sense that it adheres to a partial ordering similar to the following *example* structure:

$$\eta_1 \geq \eta_2, \eta_3 \quad \eta_2 \geq \eta_4, \quad \eta_3 \geq \eta_5, \quad \ldots$$

This ordering is model specific and always uniquely determined from the lattice structure, and it is thus difficult to perform integrations over such arbitrary orderings. However, owing to the probabilistic nature of the $\eta$ co-ordinate system, it is possible to impose the following recursive re-parameterisation:

$$\eta_1 = \delta_1$$
$$\eta_2 = \eta_1 + \delta_2$$
$$\eta_3 = \eta_1 + \delta_3$$
$$\vdots$$
$$\eta_D = \eta_{D-1} + \delta_D$$

where each $\delta_i \in [0, 1]$ for $1 \leq i \leq D$, and $\sum_{i=1}^{D} \delta_i = 1$. Geometrically speaking, $\boldsymbol{\delta} = \{\delta_i\}_{i=1}^{D}$ represents points in a $D$-simplex. We can then proceed to formally encode the lattice ordering constraints via an additional zeta matrix, resulting in $\boldsymbol{\eta} = \mathcal{Z}\boldsymbol{\delta}$, where each $\mathcal{Z}_{ij} \in \{0, 1\}$ is the value of zeta function $\zeta(q_i, q_j) = \mathbf{1}_{q_i \leq q_j}$ for the corresponding elements $q_i$ and $q_j$ in the lattice. In other words, points from the $D$ simplex, $\boldsymbol{\delta}$, can be transformed into $\boldsymbol{\eta}$ co-ordinates for the poset manifold through a linear mapping. In order to re-express the volume integral via the $\boldsymbol{\delta}$ co-ordinates, it is necessary to calculate the determinant of the Jacobian transformation matrix between the co-ordinate systems. However this is trivially one, since $\mathcal{Z}$ is by construction upper triangular, resulting in $\det\left(\frac{\partial \boldsymbol{\eta}}{\partial \boldsymbol{\delta}}\right) = 1$. Therefore it is possible to calculate the log-volume integral as,

$$\log\left(\int_{\triangle_D} \sqrt{\det\left(\mathcal{G}(\boldsymbol{\delta})\right) \cdot \det\left(\frac{\partial \boldsymbol{\eta}}{\partial \boldsymbol{\delta}}\right)^2} d\boldsymbol{\delta}\right) = \log\left(\int_{\triangle_D} \sqrt{\det\left(\mathcal{G}(\boldsymbol{\delta})\right)} d\boldsymbol{\delta}\right), \tag{3}$$

where the coordinate transformation was performed using the square of the determinant, as the metric tensor is rank (0,2) - that is, it is a doubly covariant object, and we define the $D$-simplex as $\triangle_D$. In this form it is natural to re-express the volume integral via the expectation operator, where the expectation is taken with respect to a Dirichlet distribution over $\boldsymbol{\delta}$, as the Dirichlet distribution represents a pdf over the probability simplex. In other words we consider,

$$\log\left(\int_{\triangle_D} \sqrt{\det\left(\mathcal{G}(\boldsymbol{\delta})\right)} d\boldsymbol{\delta}\right) = \log\left(\int_{\triangle_D} \sqrt{\det\left(\mathcal{G}(\boldsymbol{\delta})\right)} \cdot \frac{w(\boldsymbol{\delta})}{w(\boldsymbol{\delta})} d\boldsymbol{\delta}\right)$$
$$= \log\left(\mathbb{E}\left[\frac{\sqrt{\det\left(\mathcal{G}(\boldsymbol{\delta})\right)}}{w(\boldsymbol{\delta})}\right]\right), \tag{4}$$

where $w(\boldsymbol{\delta}) = \frac{\prod_{i=1}^{D} \Gamma(\alpha_i)}{\Gamma\left(\sum_{i=1}^{D} \alpha_i\right)} \prod_{i=1}^{D} x_i^{\alpha_i - 1} := \text{Dir}(\boldsymbol{\delta}; \boldsymbol{\alpha})$, with $\boldsymbol{\alpha} = (\alpha_1, \ldots, \alpha_D)$, and $\Gamma : \mathbb{R} \to \mathbb{R}$ being the standard Gamma function. Here, the choice of $\boldsymbol{\alpha}$ controls the manner in which sampling is performed over $\triangle_D$. We opt for a uniform exploration over the $D$-simplex, which equates to requiring that $\alpha_d = 1$ for all $d \in D$. Doing so means that we get, $w(\boldsymbol{\delta}) = \frac{\prod_{i=1}^{D} \Gamma(1)}{\Gamma\left(\sum_{i=1}^{D} 1\right)} = \frac{1}{\Gamma(D)}$, where $\Gamma(D) = (D - 1)!$. Ultimately, Equation (4) becomes,

$$\log\left(\mathbb{E}\left[\frac{\sqrt{\det\left(\mathcal{G}(\boldsymbol{\delta})\right)}}{w(\boldsymbol{\delta})}\right]\right) = \log\left(\mathbb{E}\left[\sqrt{\det\left(\mathcal{G}(\boldsymbol{\delta})\right)}\right]\right) - \log \Gamma(D), \tag{5}$$

implying that bounding the volume can be equivalently achieved by bounding the behaviour of $\log\left(\mathbb{E}\left[\sqrt{\det(\mathcal{G}(\boldsymbol{\delta}))}\right]\right)$, and then appending the $\log \Gamma(D)$ term. The upper and lower bounds on this volume integral, are shown in Appendix A.2.

## A.2 Proof of Theorem 2

In this subsection we proceed to find lower and upper bounds for the $\log V$ in the case of the prescribed lattice geometry, by exploiting the re-parameterization of the $\eta$ co-ordinate system.

*Proof.* **Volume Upper Bound:**

From Hadamarad's inequality: $|\det(A)| \leq \prod_{i=1}^{D} A_{ii}$, for $A \in \mathbb{R}^{D \times D}$. Thus:

$$|\det(\mathcal{G}(\boldsymbol{\delta}))| = \det\left(\mathcal{G}(\boldsymbol{\delta})\right) \qquad (\mathcal{G}(\boldsymbol{\delta}) \succ 0)$$

$$\leq \prod_{i=1}^{D} \mathcal{G}_{ii}(\boldsymbol{\delta}),$$

$$\Rightarrow \qquad \sqrt{\det(\mathcal{G}(\boldsymbol{\delta}))} \leq \sqrt{\prod_{i=1}^{D} \mathcal{G}_{ii}(\boldsymbol{\delta})}$$

$$\Longleftrightarrow \qquad \mathbb{E}\left[\sqrt{\det(\mathcal{G}(\boldsymbol{\delta}))}\right] \leq \mathbb{E}\left[\sqrt{\prod_{i=1}^{D} \mathcal{G}_{ii}(\boldsymbol{\delta})}\right]$$

$$\Longleftrightarrow \qquad \log\left(\mathbb{E}\left[\sqrt{\det(\mathcal{G}(\boldsymbol{\delta}))}\right]\right) \leq \log\left(\mathbb{E}\left[\sqrt{\prod_{i=1}^{D} \mathcal{G}_{ii}(\boldsymbol{\delta})}\right]\right)$$

$$\Longleftrightarrow \quad \log\left(\mathbb{E}\left[\sqrt{\det(\mathcal{G}(\boldsymbol{\delta}))}\right]\right) - \log\Gamma(D) \leq \log\left(\mathbb{E}\left[\sqrt{\prod_{i=1}^{D} \mathcal{G}_{ii}(\boldsymbol{\delta})}\right]\right) - \log\Gamma(D)$$

$$\Longleftrightarrow \qquad \log V \leq \log\left(\frac{\mathbb{E}\left[\sqrt{\prod_{i=1}^{D} \mathcal{G}_{ii}(\boldsymbol{\delta})}\right]}{\Gamma(D)}\right),$$

**Volume Lower Bound:**

$$\log\left(\mathbb{E}\left[\sqrt{\det\left(\mathcal{G}(\boldsymbol{\delta})\right)}\right]\right) \geq \mathbb{E}\left[\log\left(\sqrt{\det\left(\mathcal{G}(\boldsymbol{\delta})\right)}\right)\right] \tag{6}$$

$$= \frac{1}{2}\mathbb{E}\left[\log \circ \det\left(\mathcal{G}(\boldsymbol{\delta})\right)\right], \tag{7}$$

where Inequality (6) is Jensen's inequality. Moreover, $\mathcal{G} \succ 0 \Rightarrow \exists \mathcal{M}$ s.t. $\mathcal{G} = \mathcal{M}\mathcal{M}^{\intercal}$, where $\mathcal{M}$ is a triangular matrix (Cholesky decomposition). Thus,

$$\det\left(\mathcal{G}\right) = \det\left(\mathcal{M}\mathcal{M}^{\intercal}\right)$$
$$= \det\left(\mathcal{M}\right) \cdot \det\left(\mathcal{M}^{\intercal}\right)$$
$$= \det\left(\mathcal{M}\right) \cdot \det\left(\mathcal{M}\right)$$
$$= \det\left(\mathcal{M}\right)^2$$
$$= \left(\prod_{i=1}^{D} \mathcal{M}_{ii}\right)^2, \tag{8}$$

$$\Rightarrow \qquad \frac{1}{2}\mathbb{E}\left[\log \circ \det\left(\mathcal{G}(\boldsymbol{\delta})\right)\right] = \frac{1}{2}\mathbb{E}\left[\log\left(\prod_{i=1}^{D}\mathcal{M}_{ii}(\boldsymbol{\delta})\right)^2\right]$$

$$= \mathbb{E}\left[\sum_{i=1}^{D}\log\left(\mathcal{M}_{ii}(\boldsymbol{\delta})\right)\right],$$

$$\Rightarrow \qquad \log\left(\mathbb{E}\left[\sqrt{\det\left(\mathcal{G}(\boldsymbol{\delta})\right)}\right]\right) \geq \mathbb{E}\left[\sum_{i=1}^{D}\log\left(\mathcal{M}_{ii}(\boldsymbol{\delta})\right)\right]$$

$$\Longleftrightarrow \quad \log\left(\mathbb{E}\left[\sqrt{\det\left(\mathcal{G}(\boldsymbol{\delta})\right)}\right]\right) - \log\Gamma(D) \geq \mathbb{E}\left[\sum_{i=1}^{D}\log\left(\mathcal{M}_{ii}(\boldsymbol{\delta})\right)\right] - \log\Gamma(D)$$

$$\Longleftrightarrow \qquad \log V \geq \mathbb{E}\left[\sum_{i=1}^{D}\log\left(\mathcal{M}_{ii}(\boldsymbol{\delta})\right)\right] + \log\left(\frac{1}{\Gamma(D)}\right)$$

341 $\qquad\qquad\qquad\qquad\qquad\qquad\qquad\qquad\qquad\qquad\qquad\qquad\qquad\qquad\qquad\qquad$ $\square$

### A.3 Proof of Remark 1

343 Here we show that as $D \to \infty$, $V \to 0$ which implies that $\log V \to -\infty$. This is an important
344 indicator in the increase of generalization performance, as sufficiently large $D$ can therefore overpower
345 the $\mathcal{O}(D)$ model complexity term, present in traditional AIC and BIC.

346 *Proof.* As this represents a volume integral a trivial lower bound is zero. It follows that

$$0 \leq V \leq \mathbb{E}\left[\frac{\sqrt{\prod_{i=1}^{D}\mathcal{G}_{ii}(\boldsymbol{\delta})}}{\Gamma(D)}\right]$$

$$\Rightarrow \lim_{D\to\infty} 0 \leq \lim_{D\to\infty} V \leq \lim_{D\to\infty} \mathbb{E}\left[\frac{\sqrt{\prod_{i=1}^{D}\mathcal{G}_{ii}(\boldsymbol{\delta})}}{(D-1)!}\right].$$

347 Since each $\mathcal{G}_{ii}(\boldsymbol{\delta}) \in [0,1]$, the factorial in the denominator strongly dominates, so that:

$$\lim_{D\to\infty} \mathbb{E}\left[\frac{\sqrt{\prod_{i=1}^{D}\mathcal{G}_{ii}(\boldsymbol{\delta})}}{(D-1)!}\right] = 0.$$

348 Thus from via an application of squeeze theorem we see that,

$$0 \leq \lim_{D\to\infty} V \leq 0,$$
$$\Rightarrow \lim_{D\to\infty} V = 0.$$

349 $\qquad\qquad\qquad\qquad\qquad\qquad\qquad\qquad\qquad\qquad\qquad\qquad\qquad\qquad\qquad\qquad$ $\square$

### A.4 Visual Summary of the Effect of log$V$

351 A visual summary of the effect of the $\log V$ term in the modeling behaviour for test is made clear
352 in Figure A2b. This plot has been produced with respect to toy data, where the behaviour of the
353 log likelihood has been modeled wlog as: $\log(10^3/D^{10})$ to model rapidly increasing log likelihood
354 as $D$ increases. The $\mathcal{O}(D)$ term has been modeled in accordance to BIC as: $D/2\log N$, and the
355 value of $N$ in Figure A2b has been changed accordingly (for three different cases). Lastly, the $\log V$
356 value has been determined by constructing the metric tensor from first principles and evaluating the
357 volume integral from first Monte Carlo sampling. Firstly, it is clear that the presence of $\log V$ term
358 seems to drive a generalization behaviour that bears striking resemblance to the double-descent risk

curves. Similar intuitions were shown by Sun & Nielsen [35] for deep neural networks, which is shown in Figure A3. However, in their analysis it was necessary to consider an additional term in the MDL expansion, and instead of decreasing volume, they report having an *increasing* volume term. Thus it would appear that these higher order terms in the MDL-like expansion are highly model specific, and are required to be examined on a case-by-case basis. Indeed a blanket statement such as: *the log-volume term drives a double descent behavior* is not correct, but it can be said that for fully-observed statistical lattice models this appears to be the case. Regardless, it would appear that for a more complete understanding of the peculiarities of the *modern ML regime*, a geometric perspective seems to be certainly invaluable.

Secondly, in Figure A2b the double descent peak is observed to shift to the right with increasing $N$. Information theoretically, increasing $N$ proliferates the total number of possible encodings which may be able to explain the observed data. This is an interpretation which is consistent with Rissanen's original derivation of MDL, in which he states: "the number of distinguishable models grows with the length of the data, which seems reasonable. In view of this we define the model complexity (as seen through the data)" [27]. Interestingly, this can imply that when a model's total distinguishability is insufficient (weak regularization, and or insufficient noise), it is possible for the model to generalize well on one quantity of data, $N_1$, and then upon re-training on some new data such that $N_2 > N_1$, to then generalize poorly, due to the ability of the double-descent cusp to shift towards the right. Similar ideas have been uttered recently by Nakkiran et al., in that: "for a fixed architecture and training procedure, more data (can) actually hurt" [25]. Ultimately, such analysis seem to suggest that for a more holistic understanding on the double-descent phenomenon a geometric approach provides a more complete understanding, with invaluable intuition and assistance.

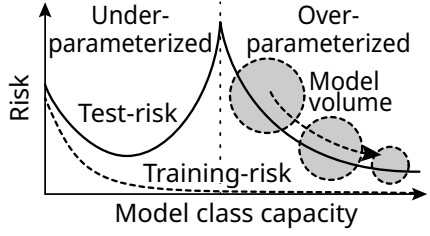

(a) Double descent risk proposed behavior for lattice-type statistical models.

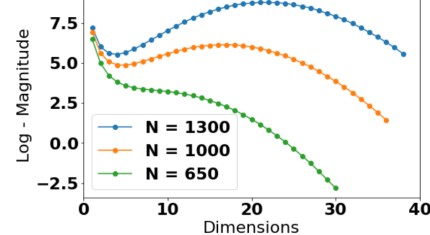

(b) Three different data cases on MDL evaluations (y-values).

Figure A2: Comparing double descent risk shapes to MDL calculations for statistical lattice machines (of which Boltzmann machines form a subset).

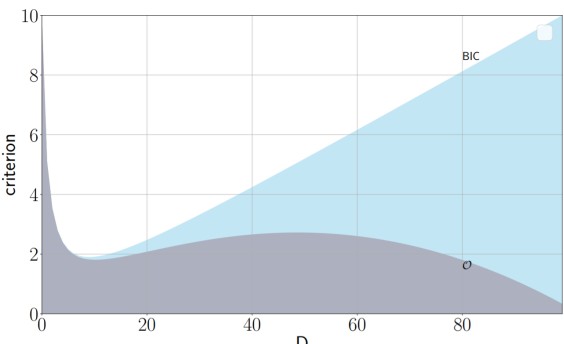

Figure A3: Deep learning razor based on inspecting the geometric structure of MDL-like expressions in Sun & Nielsen [35].

## A.5 Proof of Remark 2

In Risannen's MDL derivation for decision trees [28] (and finite alphabet processes), he derives a log-volume term as follows, $\log V = \log \frac{\pi^{D/2}}{\Gamma(D/2)}$. Since there is a topological similarity in the

structure between lattice models and tree structures (not necessarily one-to-one), it is tempting to determine under what conditions Risannen's volume expression can be derived from the poset formulations shown here. Note that in Risannen's tree structure there is no inequality symbol needed for his log-volume expression. This is because the FIM for his model is diagonal, and so there is no need to invoke Hadamarad's inequality to make the volume integral tractable (which is done here).

*Proof.* Consider an upper-bound on $\log V$, established through Hadamarad's inequality of determinants: $\sqrt{\det(\mathcal{G}(\eta))} \leq \sqrt{\prod_{i=1}^{D} \mathcal{G}(\eta)_{ii}} = \sqrt{\prod_{i=1}^{D} \eta_i(1-\eta_i)}$, which results because each diagonal term in our formulation, $\mathcal{G}(\eta)_{ii} = \eta_i(1-\eta_i)$. Thus,

$$
V = \int_{\triangle_{D-1}} \sqrt{\det(\mathcal{G}(\boldsymbol{\eta}))} d\boldsymbol{\eta}
$$

$$
\leq \int_{\triangle_{D-1}} \sqrt{\prod_{i=1}^{D} \mathcal{G}(\boldsymbol{\eta})_{ii}} d\boldsymbol{\eta}
$$

$$
= \int_{\triangle_{D-1}} \sqrt{\prod_{i=1}^{D-1} \eta_i(1-\eta_i)} d\boldsymbol{\eta}
$$

where we denote the D-1 dimensional simplex via $\triangle_{D-1}$. Our aim here is to transform the expression $\sqrt{\prod_{i=1}^{D-1} \eta_i(1-\eta_i)}$ into a more amenable term, from which the above integral can be evaluated. This can be established by expanding and collecting polynomial terms. Consider,

$$
\prod_{i=1}^{D-1} \eta_i(1-\eta_i) = \prod_{i=1}^{D-1} \eta_i(1-\eta_1)(1-\eta_2)...(1-\eta_{D-1})
$$

$$
= \prod_{i=1}^{D-1} \eta_i(1-\eta_1-\eta_2+\eta_1\eta_2)(1-\eta_3)...(1-\eta_{D-1})
$$

$$
= \prod_{i=1}^{D-1} \eta_i \left( 1 - \sum_{i=1}^{D-1}\eta_i + \sum_{i \neq j}^{D-1} \eta_i\eta_j - \sum_{i \neq j \neq k}^{D-1} \eta_i\eta_j\eta_k + ... + (-1)^D \prod_{i=1}^{D-1}\eta_i \right)
$$

$$
= \left(\prod_{i=1}^{D-1}\eta_i\right)\left(1-\sum_{i=1}^{D-1}\eta_i\right) + \left(\prod_{i=1}^{D-1}\eta_i\right)\left(\sum_{i \neq j}^{D-1}\eta_i\eta_j - \sum_{i \neq j \neq k}^{D-1}\eta_i\eta_j\eta_k + ... + (-1)^D \prod_{i=1}^{D-1}\right)
$$

where the above is a result of expanding the polynomial terms and collecting like expressions into their respective groups, and expanding the brackets. Therefore the integrand can be considered to evaluate as:

$$
\sqrt{\prod_{i=1}^{D} \mathcal{G}(\eta)_{ii}} = \sqrt{\prod_{i=1}^{D-1} \eta_i(1-\eta_i)}
$$

$$
= \sqrt{\left(\prod_{i=1}^{D-1}\eta_i\right)\left(1-\sum_{i=1}^{D-1}\eta_i\right) + \left(\prod_{i=1}^{D-1}\eta_i\right)\left(\sum_{i \neq j}^{D-1}\eta_i\eta_j - \sum_{i \neq j \neq k}^{D-1}\eta_i\eta_j\eta_k + ... + (-1)^D \prod_{i=1}^{D-1}\right)}
$$

$$
= \sqrt{\left(\prod_{i=1}^{D-1}\eta_i\right)\left(1-\sum_{i=1}^{D-1}\eta_i\right) + \mathcal{O}(\eta^3)}
$$

$$
\approx \sqrt{\left(\prod_{i=1}^{D-1}\eta_i\right)\left(1-\sum_{i=1}^{D-1}\eta_i\right)}
$$

where the approximation results from observing that each $\eta_i \in [0, 1]$. In particular, high polynomial orders of $\eta_i$ must quickly approach zero, with the speed of approach increasing as the order becomes higher. This would only not be the case if many $\eta_i = 1$. However, such a system would imply a trivial poset structure, since our partial ordering needs $\eta_i > \eta_j$ for $i > j$ for non-trivial lattice structures. In other words, since the geometric co-ordinates for the lattice structure are expressed through $\eta$, for non-trivial co-ordinate space it is required for the majority of $\eta \in (0, 1)$.

The purpose of re-writing the integrand in this way, was to motivate the usage of a Type I Dirichlet Integral for its evaluation. In particular, such integrals subscrie to the following evluation rule:

$$\int_{\triangle_D} \prod_{i=1}^{D} x^{\alpha_i - 1} f\left(\sum_{i=1}^{D} x_i\right) d^D x_i = \frac{\prod_{i=1}^{D} \Gamma(\alpha_i)}{\Gamma(\sum_{i=1}^{D} \alpha_i)} \int_0^1 f(\tau)\tau^{\sum_{i=1}^{D} \alpha_i - 1} d\tau.$$

In order to evaluation our volume integral we must consider, (i) Defining the function $f : y \mapsto \sqrt{1 - y}$, where $y := \sum_{i=1}^{D-1} \eta_i$ and (ii) Each $\alpha_i = 3/2$. These reasons are made explicit as follows:

$$\int_{\triangle_{D-1}} \prod_{i=1}^{D-1} x^{\alpha_i - 1} f\left(\sum_{i=1}^{D-1} x_i\right) d^{D-1} x_i = \int_{\triangle_{D-1}} \prod_{i=1}^{D-1} \eta^{1.5-1} f\left(\sum_{i=1}^{D-1} \eta_i\right) d\boldsymbol{\eta}$$

$$= \int_{\triangle_{D-1}} \prod_{i=1}^{D-1} \eta^{0.5} \left(1 - \sum_{i=1}^{D-1} \eta_i\right)^{0.5} d\boldsymbol{\eta}$$

$$= \int_{\triangle_{D-1}} \sqrt{\prod_{i=1}^{D-1} \eta_i(1 - \eta_i)} d\boldsymbol{\eta}$$

where evidently it can be seen that $\alpha_i = 3/2$, and $f\left(\sum_{i=1}^{D-1} \eta_i\right) = \sqrt{\left(1 - \sum_{i=1}^{D-1} \eta_i\right)}$. Therefore, we can evaluate the volume integral upper-bound as follows:

$$\int_{\triangle_{D-1}} \sqrt{\prod_{i=1}^{D-1} \eta_i(1 - \eta_i)} d\boldsymbol{\eta} = \frac{\prod_{i=1}^{D-1} \Gamma\left(\frac{3}{2}\right)}{\Gamma\left(\sum_{i=1}^{D-1} \frac{3}{2}\right)} \int_0^1 (1 - \tau)^{1/2} \tau^{\sum_{i=1}^{D-1} (3/2) - 1} d\tau \tag{9}$$

From this evaluation, it can be noticed that the new integral on the RHS is in fact an Euler-Beta integral, which evaluates according to the following rule:

$$\int_0^1 (1 - x)^{\beta - 1} x^{\alpha - 1} dx = \frac{\Gamma(\alpha)\Gamma(\beta)}{\Gamma(\alpha + \beta)},$$

where $\beta = 3/2$ and $\alpha = \sum_{i=1}^{D-1} (3/2)$ in this case. Thus:

$$\int_{\triangle_{D-1}} \sqrt{\prod_{i=1}^{D-1} \eta_i(1 - \eta_i)} d\boldsymbol{\eta} = \frac{\prod_{i=1}^{D-1} \Gamma\left(\frac{3}{2}\right)}{\Gamma\left(\sum_{i=1}^{D-1} \frac{3}{2}\right)} \cdot \frac{\Gamma\left(\frac{3}{2}\right) \Gamma\left(\sum_{i=1}^{D-1} \frac{3}{2}\right)}{\Gamma\left(\sum_{i=1}^{D-1} \frac{3}{2} + \frac{3}{2}\right)}$$

$$= \frac{\prod_{i=1}^{D} \Gamma\left(\frac{3}{2}\right)}{\Gamma\left(\sum_{i=1}^{D} \frac{3}{2}\right)}.$$

From the Legendre relation for the $\Gamma$ function, the following recurrence relation holds: $\Gamma(x)\Gamma(x + 1/2) = 2^{1-2x}\sqrt{\pi}\Gamma(2x)$. This allows one to evaluation $\Gamma(3/2)$ as follows:

$$\Gamma(1)\Gamma(3/2) = 2^{-1}\sqrt{\pi}\Gamma(2)$$
$$\Rightarrow \Gamma(3/2) = 2^{-1}\sqrt{\pi}.$$

415 Moreover, since by definition of the Gamma function we can write $\Gamma\left(\sum_{i=1}^{D} \frac{3}{2}\right) = \Gamma\left(D + \frac{D}{2}\right) =$
416 $\Gamma\left(\frac{D}{2}\right)(D-1)!$, it can therefore be established that:

$$\frac{\prod_{i=1}^{D}\Gamma\left(\frac{3}{2}\right)}{\Gamma\left(\sum_{i=1}^{D} \frac{3}{2}\right)} = \frac{\pi^{D/2}}{\Gamma(D/2)} \cdot \frac{1}{2^{D}(D-1)!},$$

417 Establishing now the $\log V$ term one can arrive at the following conclusions:

$$\log V \leq \log\left(\frac{\pi^{D/2}}{\Gamma(D/2)}\right) + \log\left(\frac{1}{2^{D}(D-1)!}\right)$$
$$= \log\left(\frac{\pi^{D/2}}{\Gamma(D/2)}\right) - \log\left(2^{D}\Gamma(D)\right)$$
$$\leq \log\left(\frac{\pi^{D/2}}{\Gamma(D/2)}\right)$$

418 Evidently, the second expression $\frac{1}{2^{D}\Gamma(D)}$ goes to zero much faster than $\frac{\pi^{D/2}}{\Gamma(D/2)}$, especially for large
419 $D$. Thus, as before we take the dominant (first) term in this approximation and arrive at:

$$\log V \leq \log\left(\frac{\pi^{D/2}}{\Gamma(D/2)}\right)$$

420 as required. □

