# OpenReview forum: "The Volume of Non-Restricted Boltzmann Machines and Their Double Descent Model Complexity"
_NeurIPS.cc/2020/Workshop/DL-IG — NeurIPSW 2020: DL-IG Oral_

### Official Review · AnonReviewer1 · 2020-10-27
**Review of "The Volume of Non-Restricted Boltzmann Machines and Their Double Descent Model Complexity"**

**Rating:** 8
**Confidence:** 4

**Review:**

This paper explores a phenomenon of intense recent interest, double descent, where for increasing model complexity we see test error falls, rises, then falls again for extremely over-parametrized models. The paper studies a particular class of lattice models for which some theoretical observations can be made. In particular, defining model complexity in a Bayesian setting, they show that in addition to the number of parameters, an effective volume of distinguishable models plays a role. Moreover, they bound this term and show that for the studied class, limits on the model volume lead to a double descent behavior.

The introduction to the problem and related work was clear and useful, and the approach taken was interesting and novel. I just got a taste for the approach from this short write-up, but it seems a promising direction I would like to learn more about. In a longer version, I would like to see more development of Eq. 1, as the appearance of the model volume term was not familiar to me. Also, my it seems that the information geometric results were used to derive bounds on the log-volume. I'm wondering if it is possible to define an even simpler class of models where the model volume can be understood more intuitively. Even if such a class did not exhibit double descent, it would help build intuition about this term.

---

### Author Response · Authors · 2020-12-12
**Poster Link**

The poster for this paper is available at:

https://drive.google.com/file/d/1DDPGpNsSaNfLuZbGszkpaVW9PFEMEc-t/view?usp=sharing

---

### Decision · Program_Chairs · 2020-11-07

Accept (Oral)